# Application of Propofol Target-Controlled Infusion for Optimized Hemodynamic Status in ESRD Patients Receiving Arteriovenous Access Surgery: A Randomized Controlled Trial

**DOI:** 10.3390/medicina58091203

**Published:** 2022-09-01

**Authors:** Po-Nien Chen, I-Cheng Lu, Tsz-Wen Huang, Po-Chun Chen, Wen-Chiao Lin, Wen-Lin Lu, Jockey Tse

**Affiliations:** 1Department of Anesthesiology, Yuan’s General Hospital, Kaohsiung 802, Taiwan; 2Department of Anesthesiology, Kaohsiung Medical University Hospital, Kaohsiung 807, Taiwan; 3Department of Anesthesiology, College of Medicine, Kaohsiung Medical University, Kaohsiung 807, Taiwan; 4Department of Anesthesiology, Kaohsiung Municipal Siaogang Hospital, Kaohsiung 812, Taiwan

**Keywords:** arteriovenous access surgery, propofol, target-controlled infusion, inhalational anesthesia, brachial plexus block, end-stage renal disease

## Abstract

*Background and Objectives*: End-stage renal disease (ESRD) is associated with increased anesthetic risks such as cardiovascular events resulting in higher perioperative mortality rates. This study investigated the perioperative and postoperative outcomes in ESRD patients receiving propofol target-controlled infusion with brachial plexus block during arteriovenous (AV) access surgery. *Materials and Methods*: We recruited fifty consecutive patients scheduled to receive AV access surgery. While all patients received general anesthesia combined with ultrasound-guided brachial plexus block, the patients were randomly assigned to one of two general anesthesia maintenance groups, with 23 receiving propofol target-controlled infusion (TCI) and 24 receiving sevoflurane inhalation. We measured perioperative mean arterial pressure (MAP), heart rate, and cardiac output and recorded postoperative pain status and adverse events in both groups. *Results*: ESRD patients receiving propofol TCI had significantly less reduction in blood pressure than those receiving sevoflurane inhalation (*p* < 0.05) during AV access surgery. Perioperative cardiac output and heart rate were similar in both groups. Both groups reported relatively low postoperative pain score and a low incidence of adverse events. *Conclusions*: Propofol TCI with brachial plexus block can be used as an effective anesthesia regimen for ESRD patients receiving AV access surgery. It can be used with less blood pressure fluctuation than inhalational anesthesia.

## 1. Introduction

End-stage renal disease (ESRD) has been associated with anesthetic risks such as cardiovascular events, anemia, bleeding diathesis, fluid imbalance, and electrolytes and acid–base disequilibrium [1]. These complications substantially increase the risk of perioperative morbidity and mortality [2,3]. However, during general anesthesia, it is challenging to maintain hemodynamic stability in ESRD patients.

Both inhalational anesthesia and total intravenous anesthesia (TIVA) are acceptable general anesthesia techniques used for ESRD patients, though volatile anesthetics tend to decrease blood pressure due to myocardial depression and reduction in systemic vascular resistance associated with their use [4]. Propofol target-controlled infusion (TCI) is widely used when performing TIVA because of its antiemetic effects, relatively rapid recovery time, and better patient satisfaction [5,6].

Because propofol does not have any analgesic properties, opioids are frequently administered by bolus dose or continuous infusion as an adjuvant in TIVA. Opioids have been associated with multiple side effects in patients with impaired renal function, and thus the search is ongoing for alternative pain relief strategies. During arteriovenous (AV) access surgery for hemodialysis, brachial plexus block has been reported to provide adequate analgesia by itself or in combination with general anesthesia [7,8]. Its use leads to sympathetic blockade, vasodilation, and improved vascular access [9,10].

Although AV access surgery can be safely performed under regional anesthesia, conversion to general anesthesia is reported in 8% of the patients receiving this surgery due to failed blockade [11]. General anesthesia may also be used from the outset if patients prefer it or if surgery involves the proximal upper arm. To date, there is no single best anesthesia technique for these patients. It is possible that combining propofol TCI and brachial plexus block might be a feasible anesthesia regimen when performing AV access surgery for ESRD patients. In our previous retrospective study, we found that propofol TCI with brachial plexus block produces more favorable hemodynamic responses than inhalational general anesthesia in ESRD patients receiving AV access surgery [12]. In order to further investigate the effect of these two anesthesia maintenance approaches in these patients, we carried out a prospective study to compare propofol TCI with a brachial plexus block and inhalation general anesthesia with brachial plexus block, on perioperative blood pressure, heart rate, and cardiac output, well as postoperative pain and perioperative adverse events.

## 2. Materials and Methods

In a preliminary trial, we found that the reduction in systolic blood pressure (mean ± standard deviation) was 12 ± 10% and 32 ± 15% in groups of patients receiving propofol TCI and sevoflurane inhalation, respectively. Before performing this study, we used a website (http://powerandsamplesize.com (accessed on 1 June 2017)) to calculate the sample size we needed for this study. Our power analysis found that a sample size of twelve patients per group could be used to obtain a power of 0.8 with an alpha error of 0.05. Therefore, we enrolled twenty-four patients in each study group. The Cohen’s d was found to be 0.788 from a calculation of the sample number of 24 based on the mean and standard deviation of blood pressure change based on a previous study [12].

### 2.1. Inclusion and Exclusion Criteria

We recruited 50 consecutive ESRD patients (age ≥ 20 years) who were diagnosed as having physical status class II-III as described by the American Society of Anesthesiologists (ASA) and who were referred to our hospital for AV access surgery in the forearm or upper arm for hemodialysis. We excluded patients who were ≥ 80 years old; had a body mass index (BMI) > 35 kg/m^2^; were allergic to propofol, sevoflurane, or local anesthetics; and had severe cardiopulmonary dysfunction, chronic pain, and dementia or severe cognitive dysfunction. The vascular access types included both AV fistula and AV graft. Each patient obtained a computer-generated random number as an allocation card. The patients were randomly divided into two groups according to anesthesia maintenance by either propofol target-controlled infusion (TCI group) or sevoflurane inhalation (General anesthesia (GA) group). An independent observer blinded to allocation collected the following information patient characteristics and medical records including gender, age, body weight, height, ASA class, history of diabetes mellitus and hypertension, type of AV access, and operation time. The primary outcome was perioperative hemodynamic stability assessed by MAP, cardiac output, and heart rate. The secondary outcomes were postoperative pain score and adverse events while the patients were in the post-anesthesia care unit (PACU). This study was approved by the institutional review board of the Kaohsiung Medical University Hospital (KMUHIRB-F(II)-20170080) and was registered with ClinicalTrials.gov (NCT03311581). Written informed consent to participate in the trial was obtained from each participant prior to surgery.

All patients received surgical procedures performed by two experienced cardiovascular surgeons. Upon arriving at the operation room, all patients received standard monitoring including electrocardiography, pulsed oximetry, and non-invasive blood pressure monitoring. The electrical cardiometry device ICON^®^ (Osypka Medical, Berlin, Germany) was used to noninvasively measure cardiac output. A standardized anesthesia induction protocol was followed using fentanyl 1 mcg/kg and propofol 2 mg/kg. A laryngeal mask airway was inserted after adequate anesthetic depth was achieved. After induction, anesthesia was maintained by either propofol TCI or sevoflurane inhalation, depending on the patient group. In the TCI group, anesthesia was maintained with propofol TCI using the Schnider model at effect-site concentration level between 1.5 and 3.0 mcg/mL. In the GA group, anesthesia was maintained with sevoflurane inhalation at the end-tidal concentration between 1.5 and 2.0%. In all patients, ultrasound-guided brachial plexus block was administered using 20 mL of local anesthesia consisting of 5 mL 0.5% lidocaine and 10 mL 0.5% ropivacaine before skin incision. Depending on the surgical site, either the interscalene or the supraclavicular approach was selected to provide appropriate dermatome coverage. In case of inadequate analgesia, defined as movement during surgery, supplemental fentanyl was administered. The intraoperative mean arterial pressure (MAP) goal was 65 mmHg. In the case that MAP was less than 50 mmHg, we first administered a bolus of ephedrine. If MAP failed to respond, we administered either dopamine, norepinephrine, or epinephrine based on clinical judgment.

The patients were transferred to the PACU after surgery. The pain score at rest and during motion was assessed using a numeric rating scale (NRS). Fentanyl 0.5 mcg/kg was given if NRS was greater than two. Adverse events in the PACU, including nausea, vomiting, itching, dizziness, hypoxia, or respiratory failure requiring mechanical support, were recorded. Postoperative pain score was also assessed post-op day 1 and post-op day 2.

### 2.2. Statistical Analysis

All data were expressed as mean (±standard deviation) or number of patients (%). Continuous variables between groups were analyzed by student *t*-test. Intragroup statistical analysis of continuous variables was performed by paired *t*-test. Categorical nominal variables were analyzed with chi-square test and Fisher’s exact test as applicable. All statistical tests were 2-tailed, and *p*-values < 0.05 were considered significant. All statistical operations were performed using Microsoft Excel 2019 (Microsoft, Redmond, WA, USA).

## 3. Results

Of the 50 patients who were initially assessed for eligibility, 1 patient was excluded because that patient did not meet the inclusion criteria and one patient did not want to participate, leaving us with 48 patients. These patients were enrolled and randomly assigned to either the TCI group (*n* = 24) or the GA group (*n* = 24). One patient in the TCI group was lost to follow-up due to missing data. We were finally left with 23 patients in the TCI group and 24 patients in the GA group, totaling 47 patients (aged between 35 and 85) whose data we analyzed (Figure 1). As can be seen in Table 1, a summary of the characteristics of the patients at baseline (sex, age, weight, height, and history of diabetes mellitus and hypertension), as well as the characteristics of their surgeries (brachial plexus block approach, AV access type and operation time), we found no significant differences between the two groups with regard to their baseline and surgical characteristics. Operation time for the TCI and GA groups ranged from 40 to 110 and 45 to 165 min, respectively.

We assessed MAP from baseline through end of surgery (Figure 2A). TCI group showed significantly less reduction in MAP than the GA group at the time of first surgical incision (*p* < 0.05) and 15 min after the first surgical incision (*p* < 0.01) and subcutaneous tunneling (*p* < 0.05) (Figure 2A). However, we found no significant difference in heart rate or cardiac output between the two groups throughout this time period (Figure 2B,C).

We found no significant difference in secondary outcome between the two groups (Table 2). The anesthesia regimen provided good surgical conditions in both groups, with only 3 (6.4%) patients moved during surgical incision. Eleven patients (23.4%) received inotropic agents during surgery. Postoperative pain intensity was relatively low in both groups, with 34 (72.3%) patients reporting an NRS ≤ 3 in the PACU. Six patients (12.8%) required rescue analgesics in the PACU. The two groups to had similar incidences of adverse events, including nausea or vomiting, pruritus, and dizziness.

## 4. Discussion

This study found the TCI group to have significantly less reduction in MAP than the GA group during AV access surgery under general anesthesia. All other perioperative and postoperative differences, including pain score and adverse events in the PACU, were insignificant.

Patients with ESRD are at greater perioperative risk of cardiovascular events, anemia, bleeding diathesis, fluid imbalance, and electrolytes and acid–base disequilibrium [1]. ESRD patients generally receive hemodialysis the day before surgery to achieve dry weight [13], so it is common that they have some hemodynamic fluctuation once general anesthesia is induced. This situation is further exacerbated by inadequate hemodynamic responses from the sympathetic nervous system, autonomic dysfunction, underlying cardiac disease, and alteration in pharmacokinetics and pharmacodynamics [1]. Compared to non-ESRD patients, patients with ESRD have been found to have inferior postoperative outcomes with a higher rate of cardiopulmonary complications, wound infection, and operation room return [14]. One recently published meta-analysis concluded that ESRD patients have 4.0 to 10.8 times the risk of postoperative mortality than non-ESRD patients [2]. Therefore, careful planning is needed to manage anesthesia in ESRD patients during surgery to ensure successful outcomes.

Most volatile agents can cause concentration-dependent decreases in myocardial contractility and systemic vascular resistance, so it is especially common for ESRD patients to encounter perioperative hypotension [15]. Therefore, the search is ongoing for alternative regimens that can achieve a better anesthesia outcome in this patient population. First approved in the late of 1990s, TCI has gained in popularity as a preferred technique used in the performance of TIVA, mostly because it makes it possible to precisely and quickly titrate plasma concentrations [16]. Pharmacokinetic studies have found that the propofol TCI model can be safely applied for use in ESRD patients [17,18]. Because it inhibits intracellular Ca2+ mobilization, propofol has been associated with preload reduction and decreases in cardiac output; however, there has been no significant difference in hemodynamic parameters between general anesthesia maintained by propofol TCI and general anesthesia maintained by sevoflurane [19]. Similar to one of our earlier studies [12], the current investigation found propofol TCI provides better hemodynamic stability than volatile anesthetics.

Like previous studies [2], we found that brachial plexus block provides excellent pain relief during AV access surgery. Only three of our patients (6.3%) moved when surgical incisions were made. Postoperatively, 13 patients (28%) patients reported moderate pain with NRS > 3, and only 6 patients (13%) needed rescue analgesics at the PACU. The result in terms of postoperative pain score is comparable to other studies [10]. We found no significant difference between the two groups with regard to surgical condition, pain intensity, and rescue analgesia use. In addition, the sensory blockade extended sufficiently into the postoperative period, resulting in less need for opioids, a great advantage for ESRD patients who usually have a higher risk of opioid-induced respiratory depression due to their impaired clearance of metabolites such as morphine-3-glucoronide and morphine-6-glucoronide [20]. Additionally, brachial plexus block also causes sympathetic blockade, resulting in vasodilation and increased blood flow, resulting in better vascular patency [9,10].

Postoperative nausea and vomiting (PONV) are important issues related to patient satisfaction with anesthesia. Prior studies and meta-analyses have shown that TIVA can be used with less PONV than inhalational anesthesia [5,21,22]. We also found the TCI group to have fewer PONV events than the GA group, though the difference was insignificant (*p* = 0.39). This reduction in significance may have been due to our small sample size or due to the fact that the brachial plexus block effectively reduces the usage of opioids, which decreases the risks of PONV.

This study has several limitations. One limitation was that we were unable to apply a double-blind strategy because differences in the performance of TIVA and inhalation anesthesia procedures were evident. Another limitation was that an objective anesthesia depth monitoring system was not routinely used on all patients. The depth of anesthesia was assessed by clinical judgment. Therefore, it is not certain whether all patients were maintained at the same depth of anesthesia. It should be added, however, that no intraoperative awareness was reported. Still another limitation is that the brachial plexus block was performed via either an interscalene or a supraclavicular approach depending on the site of the surgical incision. This might have confounded the results of the statistical analysis. This effect may be improved by using a nociception monitor to ensure that patients are under the same level of autonomic response.

## 5. Conclusions

Propofol TCI with brachial plexus block is an effective and safe anesthesia regimen for ESRD patients receiving AV access surgery. It provides less blood pressure fluctuation than inhalational anesthesia. It provides similarly satisfactory surgical conditions, low postoperative pain, and low incidence of PONV.

## Figures and Tables

**Figure 1 medicina-58-01203-f001:**
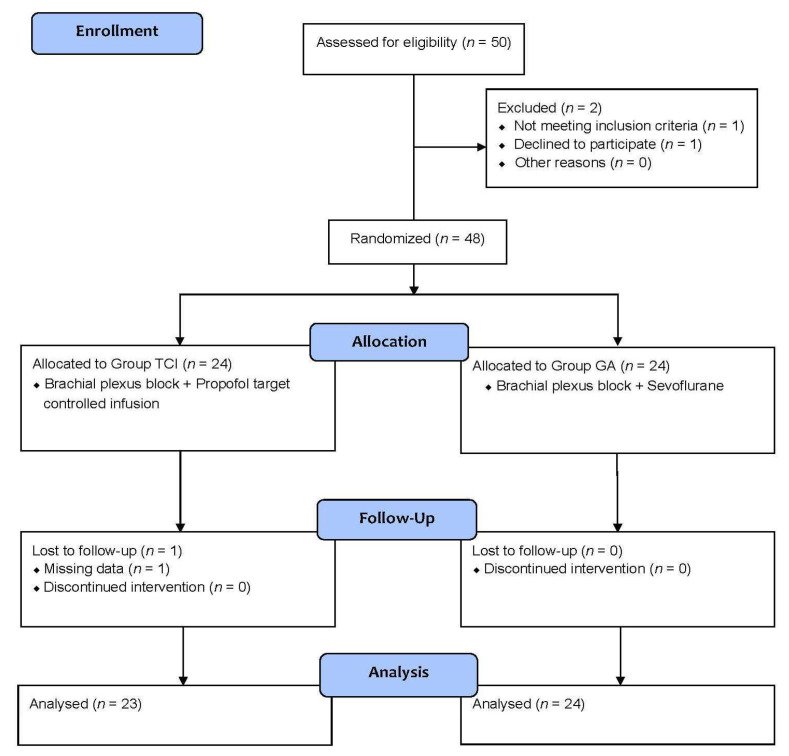
Subject enrollment flowchart. TCI = Propofol target-controlled infusion. GA = General anesthesia.

**Figure 2 medicina-58-01203-f002:**
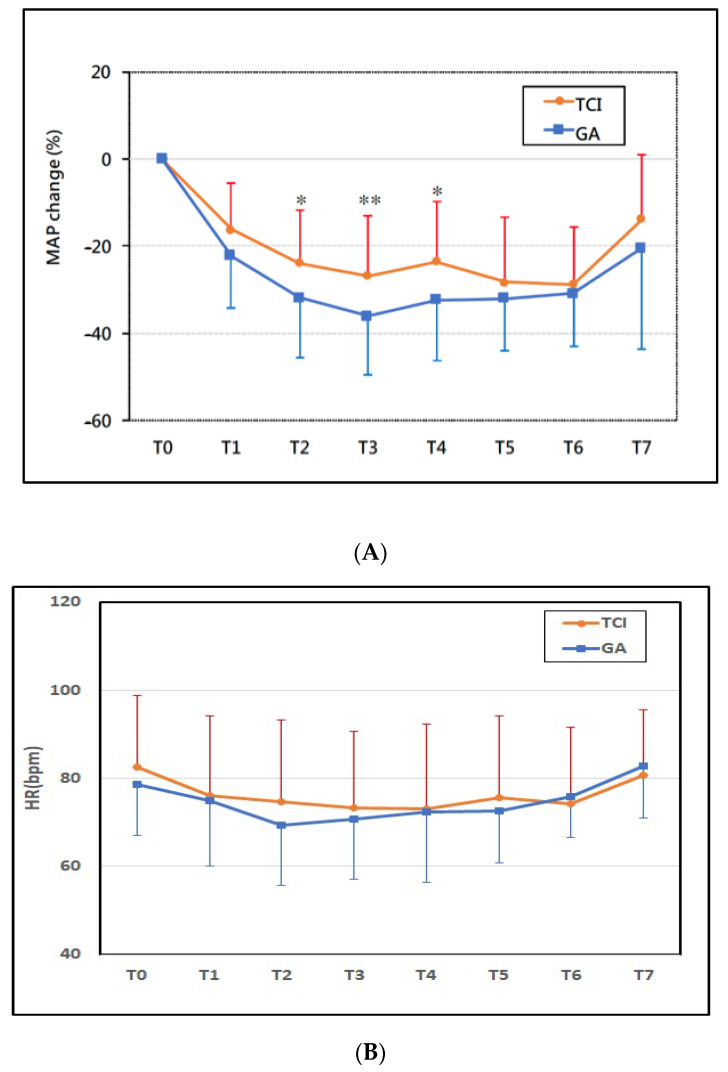
Time course of perioperative hemodynamic change. (**A**) Mean arterial pressure (MAP) change %, (**B**) heart rate (HR) bpm, and (**C**) cardiac output (CO) L/min. The TCI Group had less mean arterial pressure fluctuation than the GA group (*p* < 0.05 during T2~T4). There was no significant difference in the change in heart rate and cardiac output between groups. Time intervals were defined as follows: T0 (pre-induction), T1 (post-induction), T2 (first surgical incision), T3 (15 min after first surgical incision), T4 (subcutaneous tunneling), T5 (wound closure), T6 (the end of surgery), and T7 (arrival at Post-Anesthesia Care Unit). GA = general anesthesia with sevoflurane, TCI = propofol target-controlled infusion. * *p* < 0.05, ** *p* < 0.01.

**Table 1 medicina-58-01203-t001:** Patient characteristics.

	TCI (*n* = 23)	GA (*n* = 24)	*p*
Female/Male (n) Age (y/o) Weight (kg) Height (cm) Body mass index (kg/m^2^) Hypertension (n) Diabetes (n) Brachial plexus block (n) Interscalene/Supraclavicular Type of AV access (n) AV fistula/AV graft Operation time (min)	11/12 69.2 ± 11.3 63.5 ± 13.5 158.9 ± 7.7 25.0 ± 4.4 12 (52.2%) 10 (43.5%) 10/13 8/15 78.9 ± 16.6	13/11 64.2 ± 9.9 60.1 ± 11.3 158.3 ± 7.6 23.8 ± 3.3 15 (62.5%) 9 (37.5%) 5/19 8/16 81.4 ± 28.9	0.663 0.109 0.352 0.801 0.295 0.474 0.676 0.096 0.917 0.897

AV access = Arteriovenous access; TCI = Propofol target-controlled infusion; GA = General anesthesia.

**Table 2 medicina-58-01203-t002:** Intra-operative events and recovery profile.

Group	TCI (*n* = 23)	GA (*n* = 24)	*p*
Hemodynamic support Ephedrine Inotropics infusion * Movement during operation NRS > 3 at PACU NRS > 3 at POD 1 NRS > 3 at POD 2 Rescue analgesics at PACU Adverse events Nausea/vomiting (n) Pruritis (n) Dizziness (n) Hypoxia (n) Respiratory failure (n)	3 (13.0%) 1 (4.3%) 2 (8.7%) 6 (26.1%) 9 (39.1%) 2 (8.7%) 3 (13.0%) 1 (4.3%) 3 (13.0%) 3 (13.0%) 0 (0%) 0 (0%)	7 (29.2%) 0 (0%) 1 (4.2%) 7 (29.2%) 5 (20.8%) 7 (29.2%) 3 (12.5%) 4 (16.7%) 0 (0%) 3 (12.5%) 0 (0%) 0 (0%)	0.401 0.817 0.972 0.391 0.204 0.998 0.39 0.19 0.998 1.0 1.0

* infusion of one of dopamine, norepinephrine, or epinephrine. NRS = Numerical rating scale. PACU = Post-anesthesia care unit. POD = Post-operation day.

## Data Availability

The datasets generated and/or analyzed during the current study are available from the corresponding author upon request.

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
