# Peer review of "Application of Propofol Target-Controlled Infusion for Optimized Hemodynamic Status in ESRD Patients Receiving Arteriovenous Access Surgery: A Randomized Controlled Trial"

_medicina, 2022, doi:10.3390/medicina58091203_

Round 1

Reviewer 1 Report

The manuscript entitled “Application of Propofol Target-Controlled Infusion for optimized hemodynamic status in ESRD Patients Receiving Arteriovenous Access Surgery: A Randomized Controlled Trial” is original and has a significance for the scientific community. At the present work authors carried out a prospective study to compare propofol TCI with brachial plexus block and inhalation general anesthesia with brachial plexus block, on perioperative blood pressure, heart rate and cardiac output and well as postoperative pain and perioperative adverse events. Experimental and theoretical approach to the discussed problem is good presented in the manuscript. Obtained results are reliable and the conclusions are supported by the data collected. The manuscript is easy to read and the arguments are described in a logical and understandable way. 

As a reviewer, I have no remarks to scientific part of the investigation, however, there are some recommendation for Manuscript formalization:

1. Figure 2 – please, redraw the graphs on the Figure to make them more distinct with better quality. Then try to organize them to avoid the spare place.

2. Figure 1 – please, avoid the use of similar colours for inscription and background (for example, blue text and blue background). It would be better to improve the quality of Scheme 1 either.  

After these minor corrections, I recommend to accept this manuscript for publication.

Author Response

Dear reviewer,

Thank you very much for reviewing our article and providing useful advice.

Reviewer 2 Report

1. What was the effect size when calculating the number of samples, based on previous studies, please describe again.

2. If those allocating participants to the compared groups are aware of which group is next in the allocation process, that is, treatment or control, there is a risk that they may deliberately and purposefully intervene in the allocation of patients by preferentially allocating patients to the treatment group or to the control group and therefore this may distort the implementation of allocation process indicated by the randomization and therefore the results of the study may be distorted. Was allocation to groups concealed in this study? Please describe your research method in more detail.

3. If participants are aware of their allocation to the treatment group or to the control group  there is the risk that they may behave differently and respond or react differently to the intervention of interest or to the control intervention respectively compared to the situations when they are not aware of treatment allocation and therefore the results of the study may be distorted. Were participants blind to treatment assignment in this study? Please describe your research method in more detail.

4. If those assessing the outcomes are aware of participants’ allocation to the treatment group or to the control group there is the risk that they may behave differently with the participants from the treatment group and the participants from the control group compared to the situations when they are not aware of treatment allocation and therefore there is the risk that the measurement of the outcomes may be distorted and the results of the study may be distorted. Were outcomes assessors blind to treatment assignment in this study? Please describe your research method in more detail.

5. Please clarify the selection criteria more clearly. Because propofol and sevoflurane can act as allergens in the exclusion criteria, it is likely that subjects who are allergic to this drug should be excluded.

6. In the table, please display the p-value to the third decimal place.

7. It seems better to calculate with BMI rather than Weight (kg) and Height (cm) in the subject characteristics. For continuous variables of the subject's age and operation time (min), please provide the minimum and maximum values of interval estimates in addition to the mean and standard deviation, which are point estimates.

8. In data analysis, 'Intragroup statistical analysis of continuous variables was analyzed by paired t-test.' In the table and figure, what is the result presented as a paired t-test?

9. 'Most volatile agents can cause concentration-dependent decreases in myocardial contractility and systemic vascular resistance, so it is especially common for ESRD patients to encounter perioperative hypotension'. In this study, it is difficult to confirm that it is consistent with previous studies by presenting MAP instead of systolic and diastolic blood pressure.

10. In this study, hemodynamic variables differed only in MAP during surgery, so a description such as  'the current investigation found propofol TCI provides better hemodynamic stability than volatile anesthetics.' is considered a leap forward.

Author Response

Dear reviewer,

Please see the attachment. Thank you very much for reviewing our article and providing useful advice.

Round 2

Reviewer 2 Report

Thank you for making revision to reflect my review comments.